# HuMouS: Human Motion Synthesis with Fine-Grained Control Using Latent Space Manipulation of Cycle-Consistent Diffusion Models

## Abstract

We address the problem of spatially guided text-to-motion synthesis. While there has been work to incorporate spatial constraints in text-to-motion diffusion models, existing methods still face significant challenges in generating motions that align with the conditional controls. To this end, we propose Cycle Consistent Diffusion, a novel approach that improves controllable generation by explicitly optimizing frame-level cycle consistency between generated motions and conditional controls. Specifically, for an input conditional control, we ensure that the output motion and the input spatial constraint are forced to be consistent. A straightforward implementation though consistent with the input often does not match fine-grained control signals. To this end, we introduce a novel test-time optimization framework that directs our pre-trained cycle consistent diffusion model towards user-defined sparse constraints. We demonstrate approximately 5 to 10 percent improvement in controllability of motion synthesis on the HumanML3D dataset, while significantly reducing foot skating artifacts.

## 1 Introduction

Controlled Human Motion synthesis is essential for several applications ranging from gaming to robotics. The problem is challenging due to the immense space of possible human motions and the cost of capturing high-quality data. Recently, the emergence and improvements of diffusion models (Tevet et al., 2023b), along with the introduction of large-scale motion datasets such as AMASS (Mahmood et al., 2019) and the concomitant text-labeled motion datasets (Guo et al., 2022b) have lead to significant strides in text-to-motion generation. However, several commands cannot be entirely provided using text descriptions, and thus the provision of only text as the control signal is insufficient for several applications such as fine-grained human interaction synthesis. Often, an animator wants to provide a sparse spatial control signal along with a text input (Starke et al., 2019; Clavet, 2016). For example, an animator may wish for the precise end-effector of a character to terminate a specific location or for the character to sit at a specific location in space. In this work, we focus on the problem of incorporating spatial control signals over any joint at any given time into text-conditioned human motion generation, as shown in Fig. 1.

This problem poses significantly more challenges. While text provides an abstract signal that may be satisfied by multiple generated sequences, spatial signals provide more difficult constraints. For the objective to be adequately satisfied, the synthesized motion must match the precise spatial constraint provided by the animator, whereas such fine-grained alignment requirements are absent for text-guided synthesis. While there have been studies on incorporating spatial constraints (Xie et al., 2024; Karunratanakul et al., 2023a; Shafir et al., 2024) in diffusion-based motion synthesis methods, they either rely on approximate guidance to guide diffusion models towards motions that satisfy constraints or they require inpainting at every denoising step which in turn requires a very dense control signal. As such, their performance for sparse spatial constraints remains unsatisfactory.

To this end, we propose a novel solution that casts the problem of motion synthesis as a simultaneous sampling and optimization problem. We design a novel objective that directs spatially constrained pre-trained diffusion motion models toward satisfying user-defined sparse joint constraints. Our solution draws inspiration from ideas of test-time alignment introduced in research related to the

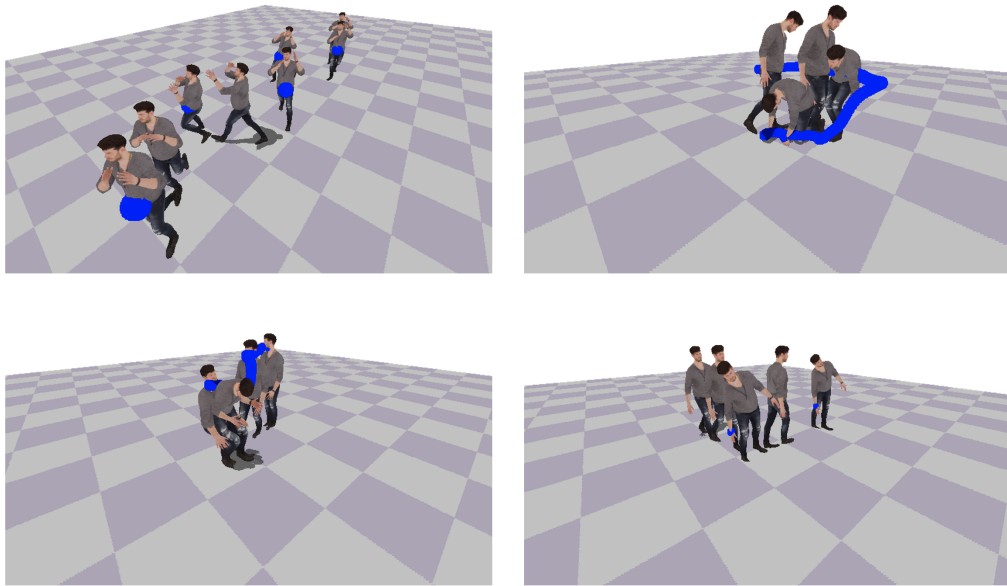

Figure 1: Given sparse spatial constraints and a text command, our method can synthesize diverse motions such as 'sit,' 'grab,' and 'crawl' and can synthesize walking in various styles while accurately following sparse spatial constraints.

sampling of text-to-image diffusion models (Prabhudesai et al., 2023; Eyring et al., 2024; Tang et al., 2024; Fan et al., 2023).

When used with existing motion diffusion architectures, such a test-time optimization often leads to degenerate solutions. To address this problem, we design a novel, Cycle Consistent, Spatially Constrained Diffusion Model that generates motions in accord with animator-provided spatial constraints. The idea is that if we translate motion from the control domain to the synthesized domain and back, we should arrive where we started. We leverage this insight to explicitly design a loss that encourages such consistency during the synthesis process.

A proper solution design adopting this idea is critical as a naive implementation typically ignores the text prompt while fully satisfying the spatial constraint, i.e., the diffusion model satisfies the reward - in our case, the spatial constraint - but ignores the original prompt. This is a common observation in diffusion sampling, called 'reward hacking' Tang et al. (2024). To address this, we introduce a novel loss function in the context of human motion that penalizes motions in the low support region of the original Gaussian noise and thus prevents reward-hacking.

Our full framework leads to $5 - 10$ percentage point improvements in terms of foot-skate ratio and control error over existing state-of-the-art spatially controlled motion synthesis methods on the HumanML3D dataset. We further demonstrate that when coupled with path planning, our idea can be used to generate long-term human motion in diverse 3D scenes. By using some user-provided spatial locations in a 3d scene as key points to direct motion, we synthesize diverse motions such as walking with raised hands and twirling chained together in 3D scenes. To the best of our knowledge, our paper is the first to demonstrate the use of diffusion models for the synthesis of chained, diverse motion with fine-grained control in 3D scenes.

To summarize, our contributions are 1) We propose a novel algorithm *HuMouS* for controlled motion synthesis that leads to state-of-the-art results in spatially constrained text-to-motion synthesis. 2) We introduce the idea of a cycle-consistent spatially constrained diffusion model for controlled motion synthesis. 3) We demonstrate that when coupled with path-planning and incorporating some sparse user-provided constraints, our framework allows for synthesizing chained diverse motions in large 3D scenes.

## 2 RELATED WORK

**Diffusion Models** Diffusion-based probabilistic generative models (DPMs) are a class of generative models learned by progressive denoising of the input data, (Ho et al., 2020; Sohl-Dickstein et al., 2015; Song & Ermon, 2019; Song et al., 2021b). Diffusion models have been successfully shown to produce state-of-the-art results in a range of diverse tasks: such as image generation (Ramesh et al., 2022; Rombach et al., 2022; Saharia et al., 2022), image-conditioned editing (Meng et al., 2022; Choi et al., 2021; Brooks et al., 2023; Hertz et al., 2022; Balaji et al., 2022), super-resolution (Saharia et al., 2021; Li et al., 2022), 3D shape generation (Poole et al., 2022; Watson et al., 2022), speech synthesis (Kong et al., 2021; Popov et al., 2021), video generation (Ho et al., 2022b;a), controlled image synthesis (Zhang et al., 2023; Ju et al., 2023) depth estimation (Saxena et al., 2023) and reinforcement learning (Janner et al., 2022). Our method is inspired by Controlnet++ (Li et al., 2024) which produces SoTA results for text-to-image synthesis by introducing the idea of cycle consistency. In contrast, our method focuses on human motion synthesis.

**Controlling Diffusion Models** Several methods have been proposed to introduce conditioning factors into the denoising process of diffusion models such as inpainting, (Chung et al., 2022; Choi et al., 2021; Meng et al., 2022), classifier-based guidance (Dhariwal & Nichol, 2021; Chung et al., 2022), and classifier-free guidance (Rombach et al., 2022; Saharia et al., 2022; Ramesh et al., 2022; Ho & Salimans, 2022). It has also been shown possible to embed images into the latent codes of the diffusion model by hacking the denoising process (Meng et al., 2022), optimizing for latent codes (Wallace et al., 2023) (Huberman-Spiegelglas et al., 2024). More recently, performing a sampling-time operation has been shown to be a powerful paradigm for synthesizing better image samples (Ben-Hamu et al., 2024; Novack et al., 2024; Tang et al., 2024).

**Human Motion Prediction.** Human Motion Prediction is a long-studied problem in vision and graphics. Early works use Hidden Markov Chains (Brand & Hertzmann, 2000) and Gaussian Processes (Wang et al., 2007), physics-based models (Liu et al., 2005) for predicting future motion. Recurrent neural networks (Graves, 2013; Hochreiter & Schmidhuber, 1997) have been used for motion prediction (Fragkiadaki et al., 2015; Martinez et al., 2017; Alahi et al., 2016) also in combination with Graph Neural Networks (Kipf & Welling; Mao et al., 2019; Li et al., 2020b; Dang et al., 2021), and variational Auto-encoders (Kingma & Welling, 2014; Habibie et al., 2017; Zhang et al., 2021; Yuan & Kitani, 2020). Transformers have recently emerged as a powerful paradigm for motion synthesis (Aksan et al., 2020; Li et al., 2021; 2020a; Petrovich et al., 2021; 2022). Motion Inbetweening (Duan et al., 2021; Harvey et al., 2020; Oreshkin et al., 2022; Yuan et al., 2022; Aksan et al., 2019; Kaufmann et al., 2020) is another classic paradigm for motion synthesis where the task is to fill in frames between animator provided keyframes. However, unlike our method, they do not focus on spatially constrained motion synthesis.

**Human Motion Synthesis.** Motion matching (Reitsma & Pollard, 2007), learned motion matching (Clavet, 2016; Holden et al., 2020) and motion graphs (Lee et al., 2002; Fang & Pollard, 2003; Kovar et al., 2008; Safonova et al., 2004; Safonova & Hodgins, 2007) are common methods employed in the video-gaming industry for generating kinematic motion sequences.

Deep learning variants such as Holden et al. (Holden et al., 2017) introduce phase-conditioning in an RNN to model the periodic nature of walking motion. In several works by Starke et al. (Starke et al., 2019; 2021; 2020), the idea of motion phases is used for motion synthesis in various settings such as a basketball game and synthetic objects. All these methods generate high-quality motion but often require manual work for non-intuitive phase labeling of phases in motion sequences. More recently (Tevet et al., 2023b), diffusion models have emerged as a powerful paradigm for human motion synthesis. Several follow-up works introduce physics (Yuan et al., 2023), blended-positional encoding (Barquero et al., 2024), field-based pose conditioning (Kulkarni et al., 2023) for improved motion quality. However, unlike our paper, they do not focus on fine-grained spatial constraints or do not condition on text. Closely related to our work, (Karunratanakul et al., 2023a) introduces the idea of optimizing latent codes of motion diffusion models, but unlike us they focus on motion editing and as our experiments indicate, their performance remains unsatisfactory for sparse-control signals.

**Humans in 3D Scenes.** The relationship between humans, scenes, and objects is another long-studied problem. Early works include methods based on 3D object detection (Gupta & Davis, 2007; Gupta et al., 2011) and affordance prediction using human poses (Delaitre et al., 2012; Grabner

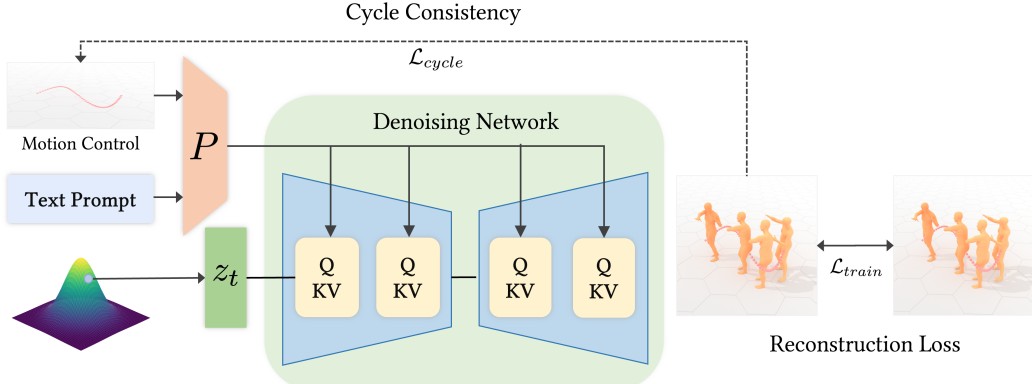

Figure 2: We train a spatially constrained diffusion by enforcing cycle consistency between the input constraint and the synthesized motion.

et al., 2011; Fouhey et al., 2014). Several recent works generate plausible static poses conditioned on a 3D scene (Li et al., 2019; Zhang et al., 2021; Wang et al., 2017; Zhang et al., 2020; Hassan et al., 2021b; Zhao et al., 2022) using recently captured human interaction datasets (Hassan et al., 2019; Guzov* et al., 2021; Savva et al., 2016; Bhatnagar et al., 2022; Taheri et al., 2020; Cao et al., 2020). Some works use reinforcement learning to synthesize walking in 3D scenes (Ling et al., 2020; Zhang & Tang, 2022; Hassan et al., 2023). Other works focus on a single action, such as grabbing or sitting (Taheri et al., 2022; Wu et al., 2022; Hassan et al., 2021a; Zhang et al., 2022) while others use VAE or mixture-of-experts networks to generate short term motion in 3D scenes. (Wang et al., 2022; 2021a; Cao et al., 2020; Wang et al., 2021b). Unlike our method, all these methods generate repetitive walking motion and do not focus on text or spatial guidance in their synthesis process.

## 3 METHOD

We aim to synthesize human motion corresponding to user-provided sparse animation signals (such as the location of the hand and the foot). To this end, we represent all motion parameters in relative coordinates (Sec. 3.1). We first train a Spatially constrained Diffusion Model (Sec. 3.2) with Cycle Consistency for Joints (Sec. 3.3. We then refine the output of this step using a novel test time refinement step (Sec. 3.4). In Sec. 3.5 we further demonstrate that such motions can be chained for the synthesis of chained diverse motions in large 3D scenes.

### 3.1 BACKGROUND

**Motion generation with diffusion model.** A diffusion probabilistic model is a generative denoising model that learns to invert a forward diffusion process. A forward diffusion process is defined as $q(\boldsymbol{x}_t|\boldsymbol{x}_0) = \mathcal{N}(\sqrt{\alpha_t}\boldsymbol{x}_0, (1 - \alpha_t)\mathbf{I})$ where $\boldsymbol{x}_0$ is a clean motion and $\boldsymbol{x}_t$ is a noisy motion at the level of $t$ defined by noise schedule $\alpha_t$. Due to the specific design of the diffusion process, the reverse diffusion denoising process $p(\boldsymbol{x}_{t-1}|\boldsymbol{x}_t, \boldsymbol{x}_0)$, which starts from pure Gaussian noise $\boldsymbol{x}_T$ generates human motion, can be approximated as.

$$p(\boldsymbol{x}_{t-1}|\boldsymbol{x}_t) = \mathcal{N}(\boldsymbol{\mu}_t, (1 - \alpha_t)\boldsymbol{I}), \tag{1}$$

where $\boldsymbol{x}_t \in \mathbb{R}^{N \times D}$ denotes the motion at the $t^{\text{th}}$ noising step and there are $T$ diffusion denoising steps in total. Following (Tevet et al., 2023b), the standard in motion synthesis is to represent motion as an array of $N$ poses stacked together, where each pose has a dimension equal to the number of joints in the skeleton used $D$ is the number of features corresponding to all joints in the frame.

The mean in each step is $\boldsymbol{\mu}_t$, which is an approximated neural network $\mathcal{D}$ that learns to predict ground-truth motion from noisy motion. $\widehat{\boldsymbol{x}_0} = \mathcal{D}(\boldsymbol{x}_t, t, \boldsymbol{c}_t; \theta)$ conditioned on the timestep $t$ and a text input $\boldsymbol{c}_t$. The text condition is passed through a clip encoder (Tevet et al., 2023b) before being concatenated with the motion sequence.

The exact $\boldsymbol{\mu}_t(\theta)$ can be computed as:

$$\boldsymbol{\mu}_t = \frac{\sqrt{\bar{\alpha}_{t-1}}\beta_t}{1-\bar{\alpha}_t}\widehat{\boldsymbol{x}_0} + \frac{\sqrt{\bar{\alpha}_t}(1-\bar{\alpha}_{t-1})}{1-\bar{\alpha}_t}\boldsymbol{x}_t \tag{2}$$

where $\beta_t = 1 - \alpha_t$ and $\bar{\alpha}_t = \prod_{s=0}^{t}\alpha_s$.

The model parameters $\boldsymbol{\theta}$ are optimized to minimize the objective

$$\mathcal{L}_{\text{train}} = \|\widehat{\boldsymbol{x}_0} - \boldsymbol{x}_0\|_2^2 \tag{3}$$

where $\boldsymbol{x}_0$ is the ground-truth human motion sequence. We denote the whole function involving all the denoising steps as $\mathcal{G}$. In essence, at test time, we have a function $\mathcal{G} : \mathbb{R}^{N \times D} \mapsto \mathbb{R}^{N \times D}$ that maps Gaussian Noise $\boldsymbol{X}_T$ to motion sequences.

While diffusion models are stochastic, there exist deterministic sampling processes that share the same marginal distribution. These processes include those defined by probability flow ODE (Song et al., 2021b) or by reformulating the diffusion process to be non-Markovian as in DDIM (Song et al., 2021a).

**Motion representation.** Following (Tevet et al., 2023a) and Guo et al. (2022b), the relative-root representation (Guo et al., 2022a) has been widely adopted for text-to-motion diffusion models. This idea represents motions as a matrix of human joint features over the motion frames with shape $N \times D$, where $D = 263$ and $N$ are the representation size and the number of motion frames, respectively. Each motion frame represents root relative rotation and velocity, root height, joint locations, velocities, rotations, and foot contact labels.

### 3.2 SPATIALLY CONSTRAINED DIFFUSION MODEL

Our goal is to train a spatially constrained diffusion model which synthesizes motion in accordance with a user-provided spatial constraint $\boldsymbol{c}_s$ and text-prompt $\boldsymbol{c}_t$. While the user is free to provide spatial constraints corresponding to any frame in the motion sequence or to any joint in any of the frames we ensure that these constraints are represented in a standard format $\boldsymbol{c}_s \in \mathbb{R}^{N \times D}$ to ensure alignment with the motion representation 3.1.

In order to modify the diffusion approximation function for it it incorporates spatial constraints $\boldsymbol{c_s}$ as well $\mathcal{D}(\boldsymbol{x}_t, t, \boldsymbol{c}_t, \boldsymbol{c}_s; \theta)$, we use a spatial module $\mathcal{P}$ which learns to parse the 3D sparse locations provided by the user. Specifically, it is a trainable copy of the Transformer encoder in the motion diffusion model that learns to enforce the spatial constraints. In addition to the spatial constraint $\boldsymbol{c}_s$, this module also takes the text constraint $\boldsymbol{c}_t$ as input.

The main transformer, instead of only using self-attention during the forward pass, unlike the original MDM formulation, also incorporates a cross-attention layer. After every self-attention layer that processes the noisy motion $\boldsymbol{x}_t$, we use a cross-attention block with the output of the spatial block $P(\boldsymbol{c_s}, \boldsymbol{c_t})$. To effectively handle the sparse control signals in time, we mask out the features at frames where there are no valid control signals,

Inspired by (Zhang et al., 2023; Xie et al., 2024), the spatial module is initialized with zeros, so that at the beginning, it has numerically insignificant output. As the training goes on, the spatial module learns the spatial constraints and adds the learned feature corrections to the corresponding layers in the motion diffusion model to amend the generated motions implicitly.

### 3.3 CYCLE CONSISTENCY FOR JOINTS

Following (Xie et al., 2024), to reduce ambiguity inherent in the local pose representation (Sec. 3.1), the spatial control signal $\boldsymbol{c}_s$ is provided in the global 3D coordinates. However, this introduces a discontinuity between the input-output spaces of the diffusion model (Sec. 3.1) We transform the output of the diffusion model from local space using a function $\mathcal{T}$ that lifts the output of the diffusion model $G(\boldsymbol{c}_s, \boldsymbol{c}_t, \boldsymbol{x_T}, t)$ from local coordinates to global coordinates, where $G(\boldsymbol{c}_s, \boldsymbol{c}_t, x_T, t)$ denotes the full function that the model performs to generate the motion $\boldsymbol{x}_0$ from random noise $\boldsymbol{x_T}$.

This operation ensures that the input constraint and the output of the diffusion model are in the same space and allows us to quantify the output further. Once transformed into global coordinate $\mathcal{T}(\mathcal{G})$

Figure 3: We optimize for the latent code of our spatially constrained diffusion model. A naive implementation often ignores the text and generates foot-skate. Hence, we use a specific initialization and regularization.

can be sub-sampled using $m_s \in [0,1]^{N \times D}$ - the mask of the user provided constraint to mask out non-controlled joints. We minimize the consistency loss between the input condition $c_s$ and the corresponding output condition (see. Fig. 3) $\hat{c}_s$ of the generated motion $\mathcal{T}(\mathcal{G}(c_s, c_t, x_T, T)$:

$$\mathcal{L}_{\text{cycle}} = \mathcal{L}(c_s, m_s \odot \mathcal{T}(\mathcal{G}(c_s, c_t, x_T, T) \tag{4}$$

However, imposing a cycle-consistent loss involving the whole diffusion process is impractical because of the spatial requirements of a GPU. Instead of randomly sampling from noise, we add noise to the training motion $x_0$, using the forward process $q(x_t|x_0)$ (Sec. 3.1), thereby explicitly disturbing the consistency between the diffusion inputs $x_0$ and their conditional spatial control $c_s$.

When the added noise is small, the original motion can be predicted $x_0$ by performing a single-step sampling on the disturbed motion sequence $x_t$ and by directly using the denoised motion $\widehat{x_0} = \mathcal{D}(c_s, c_t, x_t, t)$ to impose the cycle consistency loss:

$$\mathcal{L}_{\text{cycle}} = \mathcal{L}(c_s, m_s \odot \mathcal{T}(\mathcal{D}(c_s, c_t, x_t, t))). \tag{5}$$

Essentially, the process of adding noise destroys the consistency between the input and its condition. Then the cycle consistency loss in Eq. 4 instructs the diffusion model to generate motion that can reconstruct the consistency, thus enhancing its ability to follow the spatial constraint during generation. We find, following Li et al. (2024), that only when the timestep is less than a threshold $t_{thresh}$ is there enough information in the reconstructed motion for it to be possible to impose a cycle consistency constraint. Thus the loss is the combination of diffusion training loss and reward loss:

$$\mathcal{L}_{\text{total}} = \begin{cases} \mathcal{L}_{\text{train}} + \lambda \cdot \mathcal{L}_{\text{cycle}}, & \text{if } t \leq t_{\text{thre}}, \\ \mathcal{L}_{\text{train}}, & \text{otherwise,} \end{cases} \tag{6}$$

where $t_{thre}$ denotes the timestep threshold, which is a hyper-parameter used to determine whether a noised motion $x_t$ should be utilized for reward fine-tuning.

### 3.4 RUNTIME REFINEMENT

The spatially constrained diffusion model allows us to inject 3D sparse spatial constraints into a text-to-motion synthesis framework. However, we observe that when used as a stand-alone module, the network fails to follow the exact spatial constraint. We find that the latent space of the learned Spatially constrained diffusion model (Sec. 3.2) is smooth when the spatial constraint is fixed. This motivates performing optimization on an expressive *latent* space **z**, which provides valid motion samples when decoded (Fig. 3.4). A naive refinement task can be formulated by minimizing the following loss:

$$\mathcal{L}_{reward} = ||m_s \odot \mathcal{T}(\mathcal{G}(z, c_s, c_t, T)) - c_s||_2) \tag{7}$$

It should also be noted here that when the optimization is performed with a naive text-to-motion diffusion model without any spatial conditioning, the method produces significant foot-skating. We hypothesize that the latent space of the spatial conditioned diffusion model is fundamentally different from the latent space of a regular Motion Diffusion Model as it is significantly biased towards the conditional path provided during training. trajectories when the motion covers a long spatial extent. (See Sec. 4)

We find that when formulated as above, with random initializing, the optimization outputs motion that satisfies the reward but ignores the text. This is a known problem in sampling from diffusion models (Eyring et al., 2024; Tang et al., 2024) commonly called 'reward-hacking' where the model satisfies the optimization constraint but ignores other inputs. To address this problem, we use two key ideas:

**Initialization.** We use the output of $\mathcal{G}((c_s, c_t, x_T, T)$ embedded back into the latent space of $\mathcal{G}$, using DDIM Inversion Song et al. (2020) to initialize the refinement step. We find that setting the text to an empty string leads to significant improvement in the optimization results and, as such, do not use the original noise vector mapped $x_T$ but embed the synthesized motion back to the latent code with the text-off diffusion model. Please note that without our spatially constrained diffusion, it would be impossible to provide any dense initialization to the method, and without initialization, the method produces significant foot-skate.

**Probability Regularization.** Although this strategy provides an initialization where the spatial constraints are satisfied coarsely, the optimization still generates solutions where the input text is not precisely followed and focuses more on satisfying the explicit test-time constraint. To address this, we regularize noise vectors to remain within the high-probability region of the Gaussian distribution as follows:

$$\mathcal{L}_{reg} = \mathbb{E}_\Pi \left[ \log p_1(M_1(\Pi z)) + \log p_2(M_2(\Pi z)) \right], \tag{8}$$

where $\Pi$ is a permutation matrix and $p_1(M_1(\dot{()})$ and $p_2(M_2(\dot{()})$ are regularization functions used in High-Dimensional Statsitics Wainwright (2019); Tang et al. (2024). We find this regularization to be essential for alleviating reward hacking problems in spatially constrained motion synthesis.

The final refinement problem is thus:

$$z^* = \arg\min_z \mathcal{L}_{refine} = \mathcal{L}_{reward} + \gamma \mathcal{L}_{reg}. \tag{9}$$

This optimization is iteratively solved using gradient descent. Starting from the initialized noise, we arrive at a prediction $x$, and evaluate the criterion function $\mathcal{L}_{refine}$, then obtain the gradient $\nabla_z \mathcal{L}_{refine}$ by backpropagating through the diffusion function $\mathcal{G}$ s To obtain the desired motion, we pass the optimized noise vector through the diffusion model $x_F = \mathcal{G}(c_s, c_t, z^*)$. We denote the entire algorithm detailed above using function $\mathcal{F}$. Hence, $x_F = \mathcal{F}(c_s, c_t)$.

### 3.5 CHAINED MOTION IN 3D SCENES

In this section, we demonstrate how *HuMouS* can also be used to synthesize human motion in large 3D scenes.

**Input.** We assume that the user provides $P$ sets of action-points $\mathcal{A} = \{a_i\}_{i=1}^P$, and action-texts $\mathcal{B} = \{b_i\}_{i=1}^P$. Each text command details what action is to be performed and the keypoint details where the action is to be performed, such as "person walks while waving" or "a person sits". The actions-points are sparse - such as the location of the root (for example to indicate that the character should sit at location) or the location of the right hand (for example to indicate that a person should perform a waving action at that location).

**Separate Synthesis.** Corresponding to the $P$ sets of instructions, we first synthesize $P$ sets of motion sequences. We do so by first computing an obstacle-free path between two different action-points using the A-starHart et al. (1968) algorithm. If these paths are longer than a pre-determined length, they are further broken into waypoints.

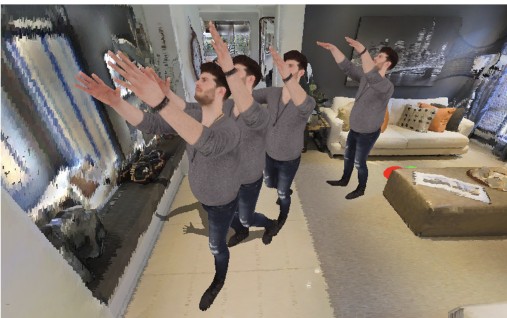

Figure 4: Our method allows for the synthesis of chained diverse motions such as dancing in 3D scenes.

These waypoints act as the sparse spatial constraint guiding the motion synthesis process. We create a spatial control singal where the root location of frames 2 seconds apart are constrained to match the waypoints. Thus corresponding to each of the $P$ instructions we define a sparse spatial constraint $\{c_s^j\}_{j=1}^P$ and $\{c_t^j\}_{j=1}^P$. Now we use these constraints with our function $\mathcal{F}$ to generate $P$ separate disjointed motion sequences - $\{s^j = \mathcal{F}(c_s^j, c_t^j)\}_{j=1}^P$ that avoid obstacles in a 3D scene and follow user-provided spatial and textual constraints.

**Chained Synthesis.** Next we describe how these disjointed motion sequences $\{s^j\}_{j=1}^P$ are joined together to form a long chained coherent motion sequence. To join sequence $j$ and $j+1$, we sample the last $Q$ frames from $s^j$ and the spatial constraint $c_s^j$ along with the first $Q$ frames from $s^{j+1}$ and the sparse spatial constraint $c_s^{j+1}$. We aim to synthesize $J = N - (2Q)$ motion frames that synthesize the transition between the two motion sequences. These two subsampled sparse spatial constraints are joined together to form an $N$ timeframe long sparse spatial constraint $c_{join}$ where the middle $J$ frames are left blank. Furthermore, since we use the SMPL parameters to represent our motion, we can define a dense spatial constraint on the $2Q$ known frames. Please note that these motion sequences are synthesized by the function $\mathcal{F}$ and hence all the SMPL, joint parameters are known. The target $c_{tar}$ thus contains joint information for every joint in the first $Q$ frames and the last $Q$ frames and is left blank for the middle $J$ frames. Using this information, we perform a refinement step that minimizes

$$\mathcal{L}_{reward} = ||\boldsymbol{m}_{tar} \odot \mathcal{T}(\mathcal{G}(\boldsymbol{c}_{join}, \boldsymbol{z}, T)) - \boldsymbol{c}_{tar}||_2) \tag{10}$$

The mask $\boldsymbol{m}_{tar}$ is defined such it is blank for the middle $J$ frames and full for the known $Q$ frames. In essence we aim to synthesize a motion sequence where the first $Q$ and last $Q$ frames match the motion synthesized in the previous step but the diffusion prior is asked to inpaint the $Q$ frames in the middle.

The steps outlined above are repeated for all the $P - 1$ transitions to finally synthesize a long chained motion sequence that respects the spatial, textual constraints defined by the user along with the constraints of the 3D scene. Please note that we do not claim to generate SoTA human motion in 3D scenes but are trying to show that diffusion allows for the synthesis of diverse chained motions in large 3d Scenes which to the best of our knowledge has not been shown before.

## 4 EXPERIMENTS

**Implementation Details.** All our experiments are done with pytorch on a single NVIDIA V100 GPU. For all experiments, we use the Adam optimizer with a decaying learning rate that starts from $10^{-5}$. For the refinement part, we use 400 steps. Our diffusion model is trained with T=1000 steps. For the refinement step, we use a deterministic DDIM-Sampler for mapping noise to motion with only 10 steps. There is a trade-off between quality and speed and we find 10 steps to be a reasonable compromise in this regard.

| | Joint | R-Precision ↑ | Diversity ↑ | Foot-Skate ↓ | Traj Err ↓ | Loc. Error ↓ | Avg Err. ↓ |
|---|---|---|---|---|---|---|---|
| Ours | | **0.724** | **9.72** | 0.0596 | **0.0389** | **0.0081** | **0.034** |
| Omnicontrol | Pelvis | 0.691 | 9.545 | **0.0571** | 0.0404 | 0.0085 | 0.0367 |
| DNO | | 0.603 | 9.345 | 0.0672 | 0.0404 | 0.0085 | 0.0389 |
| Ours | | **0.699** | **9.733** | **0.0662** | 0.0594 | 0.0094 | **0.0314** |
| Omnicontrol | Left Foot | 0.696 | 9.553 | 0.0692 | 0.0594 | 0.0094 | **0.0314** |
| DNO | | 0.603 | 9.345 | 0.0672 | **0.0404** | **0.0085** | 0.0389 |
| Ours | | **0.721** | **9.56** | **0.0648** | 0.0646 | 0.0101 | 0.0314 |
| Omnicontrol | Right Foot | 0.701 | 9.481 | 0.0668 | 0.0666 | 0.0120 | 0.0334 |
| DNO | | 0.603 | 9.345 | 0.0672 | 0.0404 | 0.0085 | 0.0389 |
| Ours | | 0.694 | **9.736** | **0.0523** | **0.0701** | **0.0114** | **0.0501** |
| Omnicontrol | Left Hand | 0.680 | 9.436 | 0.0562 | 0.0801 | 0.0134 | 0.0529 |
| DNO | | **0.712** | 9.048 | 0.069 | 0.078 | 0.0156 | 0.0558 |
| Ours | | 0.701 | **9.690** | **0.0559** | **0.0792** | **0.0121** | **0.0463** |
| Omnicontrol | Right Hand | 0.692 | 9.519 | 0.0601 | 0.0813 | 0.0127 | 0.0519 |
| DNO | | **0.768** | 9.040 | 0.0676 | 0.819 | 0.0145 | 0.0489 |
| Ours | | **0.723** | **9.233** | **0.0561** | **0.0597** | **0.0092** | **0.0371** |
| Omnicontrol | All | 0.693 | 9.016 | 0.0608 | 0.0617 | 0.0107 | 0.0404 |
| DNO | | 0.630 | 8.930 | 0.0793 | 0.0795 | 0.0011 | 0.0416 |

Table 1: Quantitative Results on the Human ML3D Dataset

**Metrics.** We adopt the evaluation protocol from (Xie et al., 2024). To evaluate and ablate our method we use the following metrics:

*R-Precision* evaluates the **relevancy** of the generated motion to its text prompt, while *Diversity* measures the **variability** within the generated motion. In order to evaluate the controlling performance, following (Karunratanakul et al., 2023b), we report *foot skating ratio* as a proxy for the **incoherence** between trajectory and human motion and **physical plausibility**. We also report *Trajectory error*, *Location error*, and *Average error* of the locations of the controlled joints in the keyframes to measure the **control accuracy**.

Following (Xie et al., 2024), all evaluations are done to generate 196 frames and five sparsity levels in the controlling signal, including 1, 2, 5, 49 (25% density), and 196 keyframes (100% density). The time steps of keyframes are randomly sampled. We report the average performance over all density levels.

**Datasets.** When applicable, we evaluate generated motions on the HumanML3D (Guo et al., 2022b) dataset, which contains 44,970 motion annotations of 14,646 motion sequences from AMASS (Mahmood et al., 2019) and HumanAct12 (Guo et al., 2020) datasets.

**Baselines.** We compare our method with the two strongest current baselines - Omnicontrol (Xie et al., 2024) and DNO (Karunratanakul et al., 2023a). Please note that as Xie et al. (2024) reports numbers for Shafir et al. (2024) that are significantly worse tha Xie et al. (2024), we do not compare with it. However, DNO focuses mainly on motion editing while we focus on controlled motion synthesis, We modify the method slightly to ensure that the comparison is fair. For initialization, we use a motion that is synthesized using MDM as there is no straightforward way to input spatial constraints to DNO. All of these existing methods use the same pose representations and thus inherit the limitations detailed in 3.1.

Our method also surpasses the previous state-of-the-art method Omnicontrol by reducing *Avg. Control err.* by 5 to 10%. In addition, our foot skating ratio is the lowest compared to all other methods.

|  | R-Precision ↑ | Diversity ↑ | Foot-Skate ↓ | Traj Err ↓ | Loc. Error ↓ | Avg Err. ↓ |
|---|---|---|---|---|---|---|
| w/o cycle | **0.724** | 9.721 | 0.0603 | **0.0389** | 0.0099 | 0.0399 |
| w/o spatial | 0.691 | 9.545 | **0.0571** | 0.0502 | 0.0125 | 0.0467 |
| w/o initialization | 0.599 | **9.733** | 0.0662 | 0.0598 | 0.0094 | 0.0384 |
| w/o regularization | 0.644 | 8.542 | 0.0601 | 0.0594 | **0.0088** | **0.0364** |
| **Full** | 0.723 | 9.233 | 0.0621 | 0.0597 | 0.0092 | 0.0371 |

Table 2: Ablation Study regarding the various components of our method.

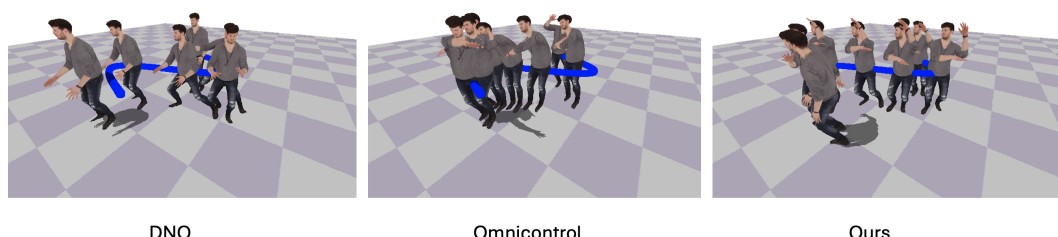

DNO        Omnicontrol        Ours

Figure 5: Text Prompt: A person walks while playing a violin. As the figure indicates, DNO often fails to obey the precise user-provided trajectory and ignores the text prompt, while Omnicontrol and DNO often produce significant foot skating artifacts. Overall, our method produces the most natural poses and follows the input prompt more closely.

## 4.1 ABLATION STUDIES

In this section we ablate the various components of our method.

There are two main components of our method: the learning part and the refinement part. In this experiment, we ablate the various components of the learning part of our method. The results of these experiments are reported in Table. 2. We switch off the cycle, and spatial encoder and do not perform any refinement. To analyze the components of our refinement step, in an experiment, we don't use any initialization, and in another one, we switch off the regularization loss. These results are reported in lines 4 and 5 of the table. As Table 2, shows all component lead to incremental improvements.

It should be noted that though the regularization and initialization increase Foot-Skate and slightly degrade the quality of control over the motion, they significantly improve the motion's fidelity to the text prompt.

## 5 CONCLUSION

We have presented a novel method for spatially constrained text-to-motion synthesis. We introduce the idea of cycle consistency in the context of human motion and show that it leads to improved performance. We also introduce the idea of latent space manipulation with a novel test-time optimization algorithm that directs pre-trained spatially constrained diffusion models toward user-defined preferences. We have further demonstrated that when coupled with path planning and some user-provided sparse key points, our framework can synthesize long-term human motion in 3D scenes. We hope our work will inspire further research in the field of text-to-motion synthesis and contribute to advancements in computer animation.

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
