# OpenReview forum: "HuMouS: Human Motion Synthesis with Fine-Grained Control using Latent Space Manipulation of Cycle-Consistent Diffusion Models"
_ICLR.cc/2025/Conference — Submitted to ICLR 2025_

### Official Review · Reviewer_hQNk · 2024-10-31

**Soundness:** 2
**Presentation:** 2
**Contribution:** 2
**Rating:** 3
**Confidence:** 4

**Summary:**

This paper presents an effort towards controllable motion generation. A "cycle-consistent spatially constraint" is proposed during motion diffusion model training. Test-time optimization is also used to enforce joint position controls. Experiments show the proposed method surpasses Omnicontrol and DNO in HumanML3D dataset.

**Strengths:**

The literature review in related works section is thorough.

**Weaknesses:**

- The writing in the method section needs to be improved. There are a lot of typos, mismatched symbols, wrong hyperlinks. To name a few, eq7 has an extra right parenthesis. Line 320, the hyperlink of "Fig. 3.4" links to subsection 3.4 instead of a figure.
- The ablation study fails to prove the effectiveness of the proposed method, where the full model lead at none of the metric.
- The evaluation and analysis is very limited. The experiment section only takes 2 pages. The qualitative evaluation is also inadequate.

**Questions:**

See the weakness section.

---

### Official Review · Reviewer_3E5X · 2024-10-31

**Soundness:** 3
**Presentation:** 2
**Contribution:** 2
**Rating:** 5
**Confidence:** 2

**Summary:**

This paper proposes Cycle Consistent Diffusion, a novel approach that improves controllable generation by explicitly optimizing frame-level cycle consistency between generated motions and conditional controls. The authors introduce a cycle consistency loss that improves the alignment of generated motions with these constraints, effectively addressing issues such as foot-skating. Additionally, this method enables the synthesis of diverse, chained motions within large 3D scenes, showcasing the framework's capability to generate coherent human movements while adhering to specified spatial and textual requirements.

**Strengths:**

1：The cycle consistency mechanism of this method provides researchers with a novel perspective, advancing the exploration of controllable generative models in directions such as multimodal data processing.

2：This method enables the generation of diverse chained motions in large 3D scenes, thereby facilitating further advancements in HSI and having strong practical applicability.

**Weaknesses:**

1：The experiments fail to demonstrate the advantage of Cycle Consistency and lack metrics to indicate the quality of the generated motion.

2: This paper proposes that the cycle-consistent loss will lead to a huge amount of computation, and the method of adding local noise and single-step sampling will reduce the burden on the GPU. But it does not show the effect after improvement, nor does it indicate whether it will affect the quality of the generated motion.

3: The experimental section of this paper lacks sufficient comparative methods, and the evaluation metrics are inadequate. Ablation studies also fails to demonstrate the effectiveness of the proposed method. Additionally. Table 1 has detailed issues such as data mislabeling

**Questions:**

1：Table 2 shows that the performance of the model without cycle consistency (w/o cycle) is less than one percentage point lower than that of the full model in terms of Loc. Error and Avg Err. However, the w/o cycle model significantly outperforms the full model in R-Precision, diversity, and foot-skate metrics. where does the advantage of cycle consistency lie? More experiments are needed to demonstrate the effectiveness of cycle consistency.

2：It is also necessary to assess the overall quality of the generated motions using additional metrics, such as FID.

3：The paper suggests that incorporating cycle consistency loss introduces substantial computational overhead, and mitigates GPU burden by using localized noise addition and single-step sampling. Please demonstrate that these methods do not compromise the quality of the generated motions, and specify the exact training and inference times in the Implementation Details. Clarification on how these decisions are algorithmically determined would enhance the transparency and reproducibility of the research.

4：Please note certain details, such as in Table 1, where the DNO model's Trajectory Error and Location Error in "Fight Foot" are significantly lower than those of ours, yet the best results are not bolded.

---

### Official Review · Reviewer_4NaM · 2024-11-03

**Soundness:** 3
**Presentation:** 2
**Contribution:** 2
**Rating:** 5
**Confidence:** 4

**Summary:**

This paper introduces a novel approach for spatially guided text-to-motion synthesis, named "HuMouS". It addresses the challenge of generating human motions that align with both textual descriptions and spatial constraints. The authors propose a method called Cycle Consistent Diffusion, which optimizes frame-level cycle consistency between generated motions and conditional controls. They also introduce a test-time optimization framework to direct a pre-trained cycle consistent diffusion model towards user-defined sparse constraints. The paper demonstrates improvements in controllability and a reduction in artifacts such as foot skating. However, I have several concerns about this paper.

**Strengths:**

1. The introduced cycle consistency mechanism is well-designed for the motion domain. An effective test-time optimization framework is proposed with proper motivation. The authors also use a novel loss function to address the reward hacking issue.
2. The experiment on the HumanML3D dataset shows that HuMouS achieves a clear improvement in quantitative metrics.

**Weaknesses:**

1. Figures do not contain much information.  Fig.2 and Fig.3 are very similar, making it difficult to understand the methods described in the paper.
2. The qualitative results are not comprehensive.  It is not easy to observe the improvement beyond DNO and Omnicontrol in the provided images. In addition, the presented figures do not provide sufficient evidence to demonstrate the effectiveness of the proposed method.
3. The ablation study reveals that incorporating all proposed components does not yield optimal performance across all metrics. Instead, the absence of certain components leads to better performance on some metrics. This raises concerns about whether the performance improvements stem from parameter tuning rather than the effectiveness of the proposed components.
4. The experimental evaluation is insufficient. The experiments are conducted on only a single dataset, lacking cross-dataset validation on other relevant datasets such as the KIT-ML dataset. Moreover, the comparison is limited to only two baseline methods, which is inadequate compared to comprehensive evaluations like those in Omnicontrol which include multiple comparative approaches.
5. Lack of analysis of failed cases and some insightful observations.

**Questions:**

1. Could you elaborate on the mechanism by which cycle consistency is enforced between spatial constraints and generated motions within your framework? Specifically, how is the interplay between these two elements managed to ensure consistency across different cycles?
2. How is your method designed to maintain temporal dependencies during the optimization process? What strategies are employed to preserve the sequential coherence of motions despite potential disruptions or variations in the data?
3. How does your method fare when applied to complex environments? What measures are taken to ensure robustness against uncertainties and variabilities typically encountered in real-world scenarios, and how effectively does the method handle these challenges?

---

### Official Review · Reviewer_99Yk · 2024-11-04

**Soundness:** 2
**Presentation:** 2
**Contribution:** 2
**Rating:** 3
**Confidence:** 4

**Summary:**

This work is trying to address the problem of motion generation with spatial control. It proposes a cycle consistency loss when training the diffusion model with spatial conditions. This is achieved by extracting the spatial control signal from the diffusion prediction and forcing an alignment with the input condition spatial signal. Furthermore, it proposes to optimize the latent noise of the diffusion model using the DDIM solver to ensure the precision of the spatial control at test time.

**Strengths:**

1. The paper adopts a cycle consistency loss to enhance the spatial control ability of the diffusion model which is new in the motion community.
2. The proposed probability regularization of the diffusion model's latent noise is new to me.
3. The paper describes the application of chained motion generation in 3D scenes in detail.
4. It shows better performance compared with the SOTA methods.

**Weaknesses:**

1. The technical novelty is relatively limited.
    1) A spatial condition motion diffusion model is very straightforward and has been seen in previous work [1].
    2) The cycle consistency training is very similar to the method used in image generation [2], which is easy to inject into the motion diffusion model without a specific design. I don't think it introduces much insight to the motion generation community.
    3) Optimizing the diffusion noise has been seen in previous work [3]. The modification is rather incremental. More analysis is needed to demonstrate the design of special DDIM inversion initialization and regularization.
2. There only limited video results are for me to evaluate the quality of the generated motion. More visual results compared with the SOTA methods are helpful for evaluation.

[1] Yiming Xie, Varun Jampani, Lei Zhong, Deqing Sun, and Huaizu Jiang. Omnicontrol: Control
any joint at any time for human motion generation. In The Twelfth International Conference on
Learning Representations, 2024.
[2] Ming Li, Taojiannan Yang, Huafeng Kuang, Jie Wu, Zhaoning Wang, Xuefeng Xiao, and Chen
Chen. Controlnet++: Improving conditional controls with efficient consistency feedback. In
European Conference on Computer Vision, 2024.
[3] Korrawe Karunratanakul, Konpat Preechakul, Emre Aksan, Thabo Beeler, Supasorn Suwa-
janakorn, and Siyu Tang. Optimizing diffusion noise can serve as universal motion priors. In
arxiv:2312.11994, 2023a.

**Questions:**

1. I suggest the author clarify their contributions compared with the existing work, such as Omnicontrol [1], GMD [3] and ODN [2].
2. I wonder if it's possible to show the effects of  Probability Regularization and different initialization using a 2D map like t-SNE.
[1] Yiming Xie, Varun Jampani, Lei Zhong, Deqing Sun, and Huaizu Jiang. Omnicontrol: Control
any joint at any time for human motion generation. In The Twelfth International Conference on
Learning Representations, 2024.
[2] Korrawe Karunratanakul, Konpat Preechakul, Emre Aksan, Thabo Beeler, Supasorn Suwa-
janakorn, and Siyu Tang. Optimizing diffusion noise can serve as universal motion priors. In
arxiv:2312.11994, 2023a.
[3] Karunratanakul, Korrawe, et al. "Guided motion diffusion for controllable human motion synthesis." Proceedings of the IEEE/CVF International Conference on Computer Vision. 2023.

---

### Meta-Review · Area_Chair_RrcP · 2024-12-17

**Metareview:**

This paper introduces Cycle Consistent Diffusion for generating human motions that align with both spatial constraints and textual descriptions. The method incorporates a cycle consistency loss during the training of a diffusion model. It also employs test-time optimization to refine motion generation for spatial control. The framework effectively addresses issues like foot-skating and enables the synthesis of diverse, spatially guided motions within large 3D scenes. Experiments demonstrate superior performance compared to baselines on the HumanML3D dataset.

After the review process, all four reviewers leaned toward rejection. The weaknesses of this submission include limited insights into applying cycle consistency to the motion diffusion model, minimal contributions, and insufficient evaluation.

**Additional Comments On Reviewer Discussion:**

After the authors' rebuttal, two reviewers participated in the discussion. However, neither was convinced by the rebuttal.

---

### Decision · Program_Chairs · 2025-01-22

Reject